# The Mammalian Cysteine Protease Legumain in Health and Disease

**DOI:** 10.3390/ijms232415983

**Published:** 2022-12-15

**Authors:** Rigmor Solberg, Ngoc Nguyen Lunde, Karl Martin Forbord, Meshail Okla, Moustapha Kassem, Abbas Jafari

**Affiliations:** 1Section for Pharmacology and Pharmaceutical Biosciences, Department of Pharmacy, University of Oslo, N-0316 Oslo, Norway; 2Department of Endocrinology and Metabolism, Odense University Hospital, University of Southern Denmark, DK-5000 Odense, Denmark; 3Department of Community Health Sciences, College of Applied Medical Sciences, King Saud University, Riyadh 12372, Saudi Arabia; 4Department of Cellular and Molecular Medicine, University of Copenhagen, DK-2200 Copenhagen, Denmark

**Keywords:** asaparginyl endopeptidae (AEP), asparaginyl carboksypeptidase (ACP), cysteine protease, δ-secretase, legumain, peptide asparaginyl ligase (PAL), prolegumain

## Abstract

The cysteine protease legumain (also known as asparaginyl endopeptidase or δ-secretase) is the only known mammalian asparaginyl endopeptidase and is primarily localized to the endolysosomal system, although it is also found extracellularly as a secreted protein. Legumain is involved in the regulation of diverse biological processes and tissue homeostasis, and in the pathogenesis of various malignant and nonmalignant diseases. In addition to its proteolytic activity that leads to the degradation or activation of different substrates, legumain has also been shown to have a nonproteolytic ligase function. This review summarizes the current knowledge about legumain functions in health and disease, including kidney homeostasis, hematopoietic homeostasis, bone remodeling, cardiovascular and cerebrovascular diseases, fibrosis, aging and senescence, neurodegenerative diseases and cancer. In addition, this review addresses the effects of some marketed drugs on legumain. Expanding our knowledge on legumain will delineate the importance of this enzyme in regulating physiological processes and disease conditions.

## 1. Introduction

### 1.1. Legumain Expression and Synthesis

Mammalian legumain (also known as asparaginyl endopeptidase or δ-secretase) is an endolysosomal cysteine protease that is highly expressed in different tissues such as kidneys, liver, spleen, placenta, and testis. Legumain was first identified in common beans (legumes, indicated by the name) in the early 1980s [1,2]. Mammalian legumain was first identified in 1996 as a putative protease [3] and characterized in 1997 [4] (reviewed in [5]). Functional human legumain is encoded by the *LGMN* gene located on chromosome 14, locus 14q32.1, giving rise to prolegumain of 433 amino acids (56 kDa) [4,6]. 

Similar to other cysteine proteases, prolegumain is synthesized as an inactive zymogen to prevent unwanted substrate proteolysis. After synthesis, prolegumain is predominantly transported via the Golgi apparatus to the endolysosomal compartments [4]. During transport, prolegumain is post-translationally modified and N-glycosylation is shown to be important for transport, cellular localization, secretion, and the stability of legumain [7]. In addition, the post-translational conjugation of ubiquitin monomers also promote prolegumain stability [8]. In the acidic and reductive endolysosomal environment, prolegumain is processed to the mature active 36 kDa form [9,10].

### 1.2. Legumain Structure and Activity

The crystal structure of human prolegumain was determined by Dall and Brandstetter in 2013 [11], revealing a catalytic domain and a carboxy-terminal prodomain containing an activation peptide (AP) and a legumain stabilization and activity-modulating (LSAM) domain covering the catalytic active site (Figure 1). The proteolytic activity of legumain is dependent on a cysteine residue situated in the active site (Cys189) of the catalytic domain, acting as a nucleophile in the hydrolytic cleavage of peptide bonds [12], thus legumain is classified as a cysteine protease [13]. Based on the topology and sequence homology, the MEROPS database classification system places legumain together with the caspases in the CD clan of proteases, thus indicating an evolutionary relationship [13]. Similar to other cysteine proteases in clan CD, legumain has a conserved motif in the active site and a strict specificity for one particular amino acid at the substrate P1 position. Legumain cleaves its substrates (Table 1) preferentially after asparagine (Asn) residues in nonterminal regions with high specificity at pH ~ 5.8, hence the synonym asparaginyl endopeptidase (AEP) is used. In contrast, the caspases prefer aspartate (Asp) in the substrate P1 position, and in more acidic conditions (pH < 4) legumain achieves caspase-like proteolytic activity [14,15]. Thus, the proteolytic activity of legumain is pH-dependent. 

The activity and stability of legumain is highly regulated by the environment. Prolegumain is stable at a neutral pH (pH 7) and forms homodimers for stabilization. Mature active legumain exist as a monomer and is rapidly and irreversibly inactivated through denaturation at a pH > 6 due to a high density of acidic amino acids that are partially protonated at acidic pH, but are negatively charged and electrostatically repulsive at near-neutral pH [11]. 

At neutral pH, the positively charged LSAM domain acts as a direct stabilizer of the catalytic domain, as it sterically shelters the active site of the negatively charged catalytic domain through electrostatic interactions, restricting proteolytic activity and ensuring prolegumain stability. Activation of the AEP activity of legumain includes several steps. During the first step, the covalent connection between the AP and LSAM is autocatalytically cleaved at a pH < 5.5, resulting in a 47 kDa intermediate product (Figure 1). The 47 kDa intermediate legumain has no AEP activity, but has been demonstrated to have asparaginyl-specific carboxypeptidase activity (ACP) at a near-neutral pH [11]. The LSAM domain, which remains electrostatically bound to the catalytic domain at a pH > 4, is important for the ACP activity. In the second step, another autocatalytic hydrolysis at a pH < 4 results in a 46 kDa intermediate active form of legumain. The decrease in the pH causes protonation to the point where the LSAM domain no longer remains electrostatically attached to the catalytic domain and the 46 kDa legumain acquires AEP activity. Although legumain is considered as an asparaginyl-specific endopeptidase, protonation of the catalytic domain at a pH < 4 also results in a conformational change known as superactivation [15]. Superactivated legumain acquires aspartyl-peptidase activity similar to the caspases, allowing for autocatalytic cleavage after Asp303/309. The amino terminus is also cleaved after Asp21/25 in this step, although this has been proven not to be essential for the AEP activation of legumain [15]. In contrast, the last step is not autocatalytic, but requires possessing after the Lys-Arg-Lys289 (KRK289) by unidentified protease(s) at a pH < 4 [10], resulting in the mature active legumain of 36 kDa (Figure 1).

Endogenous cystatins also play a role in regulating legumain activity and stability. Cystatins inhibit the AEP activity of legumain through direct binding to the active site. In addition, legumain stability is also regulated by cystatins, through complex formation with amyloid fibrils of endogenous cystatins such as cystatin E/M [16]. It is possible that cystatin amyloid fibrils serve as a binding platform for legumain, thereby stabilizing the pH-sensitive legumain [16].

Another group of environment-related factors that affect legumain activity are polyanionic glycosaminoglycans (GAGs), such as chondroitin 4-sulphate, which are among the components of the extracellular matrix in many tissues [17]. Presence of GAGs accelerate the autocatalytic activation of prolegumain through ionic interactions in a concentration-, pH-, and time-dependent manner [17].

Importantly, it was recently shown that legumain activity is also regulated by the extended active site residues, which indicate a favorable or inhibitory effect on the AEP activity of legumain [18]. In this regard, Glu190 is shown to inhibit AEP activity, whereas Ser215 and Asn42 indicate a favoring effect on the AEP activity of legumain [18]. In addition, it was also reported recently that the conformational flexibility of the substrate plays a role in defining whether it will be ligated or hydrolyzed by legumain [18].

### 1.3. Regulation of Legumain Transcription

Transcriptional regulation is another level where legumain is regulated. However, the transcription factors regulating legumain expression are only partially understood. Here, we discuss the currently known transcription factors that regulate legumain expression in the context of normal physiology or pathophysiology.

Signal transducer and activator of transcription 3 (STAT3) is involved in a wide range of biological and pathological processes through the regulation of different gene programs and mediating the communication between various cytokines and kinases [19]. STAT3 is one of the key transcription factors involved in the regulation of lysosomal proteolytic capacity [20]. It is shown that lysosomal stress due to protease/substrate imbalance leads to STAT3 activation and the subconsequent de novo expression of multiple lysosomal hydrolases, including legumain, thereby enabling lysosomal adaptation [20]. In addition, legumain deficiency has been shown to cause STAT3 activation and hyperproliferation in the spleen and kidney, which can explain the splenomegaly and hyperplasia of the proximal tubular cells observed in legumain-deficient mice [20].

The CCAAT-enhancer-binding protein beta (C/EBPꞵ) is a transcription factor involved in diverse biological processes such as adipocyte differentiation, liver regeneration, myelopoiesis, and immune response [21]. C/EBPꞵ has been shown to regulate legumain expression in the brain in an age-dependent manner, thereby contributing to the pathogenesis of Alzheimer’s disease [22,23].

The p53 tumor suppressor is a substrate of legumain (Table 1). However, it is shown that p53 regulates legumain expression, as there is a p53-binding site in intron 1 of the human and mouse legumain gene. Inducing p53 expression by doxorubicin increases legumain expression, whereas p53 knockdown reduces legumain expression in HCT116 human colon cancer cells [24,25]. In addition, DJ-1, which is known to function as a coactivator of different transcription factors, including p53, binds to the p53-binding site in intron 1 of the mouse legumain gene and exerts an inhibitory effect on legumain expression in mouse fibroblasts. In this regard, it is shown that legumain expression is increased in fibroblasts from DJ-1 knockout mice as compared to wild-type controls [25].

In addition to the transcription factors mentioned above, a functional legumain pseudogene (*LGMNP1*) also plays a role in regulation of legumain expression. The *LGMNP1* is located on human chromosome 13 and is shown to induce *LGMN* expression by antagonizing the miR-495-3p which targets legumain [26]. Increased legumain expression by *LGMNP1* is shown to be involved in tumor progression of glioblastoma [26].

**Table 1 ijms-23-15983-t001:** Known mammalian AEP substrates of legumain (alphabetical order).

Legumain Substrate	Result of Substrate Cleavage by Legumain	Reference
Acetoacetyl-CoA synthetase	Degradation	[27]
A disintegrin and metalloproteinase (ADAM) 10 and 17	Degradation	[28]
Alpha-1-macroglobulin	Activation	[29]
Alpha (α)-synuclein	Processing	[30,31]
Amphiphysin I	Processing	[32]
Amyloid precursor protein (APP)	Processing	[33]
Annexin A2	Degradation	[34,35]
Beta-amyloid protein 1–40	n/a	[29]
Betaine-homocysteine S-methyltransferase 1	Degradation	[36]
Cathepsin B, H, L	Processing	[37]
Cofilin-1	Processing	[38]
Cystatin C and E/M	Processing	[39]
Fibronectin	Degradation	[40,41]
Forkhead box P3 (FOXP3)	Degradation	[42]
Inhibitor 2 of protein phosphatase 2 (I2PP2A)/SET	Degradation	[43,44]
Invariant chain chaperone (li)	Processing	[45]
Myelin basic protein (MBP)	Processing/degradation	[46,47]
p53	Degradation	[48]
Promatrix metalloproteinase 2 (proMMP-2)	Activation	[49]
Prolegumain	Autoactivation	[9,14,15]
Prothymosin α	Processing	[50]
Serotransferrin	n/a	[29]
Serum albumin	n/a	[29]
Serin/treonin protein kinase 2 (SRPK2)	Activation	[51]
Synaptojanin 1 (SYNJ1)	Processing	[52]
TAR DNA-binding protein 43 (TDP-43)	Processing	[53]
Tau	Processing	[51,54,55,56,57]
Toll-like receptor 7 (TLR7)	Processing	[58]
Toll-like receptor 9 (TLR9)	Processing	[59,60,61]
Tropomodulin-3 (Tmod-3)	Processing	[62]
Vitamin-D-binding protein (VDBP)	Degradation	[35]

### 1.4. Legumain Cellular Localization and Substrates

Although legumain is mainly located in the endolysosomes, both pro- and active legumain have been detected in the nucleus and cytosol of colorectal cancer cells [63,64]. In addition, prolegumain has been found in the secretome of several cell types [63,65,66,67,68,69,70], as well as in body fluids [41,71,72,73,74]. In addition, legumain has been observed locally on cell surfaces and/or in extracellular matrixes [75,76,77], as well as in extracellular vesicles (EVs) [70,78]. 

The function of extracellular legumain is not fully understood. It is possible that extracellular legumain contributes to communication between different cells and tissues, as secreted prolegumain can be internalized and subsequently processed to the active form by several cell types [5,9,79]. Legumain interaction with integrin αvβ3 receptors on cell membranes stabilizes and increases the optimal pH of the AEP activity, substantiating the possibility of extracellular proteolytic activity [13,80,81,82,83,84,85,86,87,88,89,90]. Moreover, legumain has a receptor modulating effect on integrin αvβ3, altering the downstream signaling cascades in vascular smooth muscle cells [79].

An increasing number of proteins have been described to be cleaved by legumain. A list of currently known legumain (AEP) substrates is presented in Table 1. Substrate cleavage by legumain mediates either the activation, processing, or degradation of the cleaved protein. As an example, the legumain cleavage of promatrix metalloproteinase 2 (MMP-2) mediates MMP-2 activation [27]. Legumain is also responsible for processing of the lysosomal cysteine cathepsins B, H, and L from single to two-chain active forms [28]. Substrate degradation by legumain is the more common result, exemplified by fibronectin cleavage and degradation, which is shown to control extracellular matrix (ECM) remodeling [29,30].

### 1.5. Endogenous Legumain Inhibitors

To protect cells and tissues from harmful proteolytic activity, proteases are regulated by endogenous inhibitors. A subset of endogenous cysteine protease inhibitors, the type 2 cystatins, are potent legumain inhibitors. Cystatin C, E/M, and F inhibit the AEP activity of legumain, although cystatin E/M is the most potent (Ki = 1.6 pM) [80]. These cystatins bind to the catalytic site of legumain in a substrate-like manner through an asparagine residue (Asn39) in the legumain-specific exosite loop [39] (Table 1). Interestingly, the enzyme–inhibitor complex is stable through electrostatic interactions and remains stable at a pH > 6. However, intriguingly, a shift from a pH of 4 to a pH > 6 results in the religation of the cleaved cystatin peptide bond at Asn39 by legumain, demonstrating a pH-dependent equilibrium between the asparaginyl endopeptidase (AEP) and peptide asparaginyl ligase (PAL, also called butylase-1 [81]) activity of legumain. Furthermore, the 47 kDa intermediate form of legumain can be autoligated back to the original zymogen of 56 kDa at a neutral pH [82]. The ligase activity of legumain has only been demonstrated for cystatin E/M and the rezymogenization of the protease itself, both instances resulting in attenuation of the protease activity and stabilization of the catalytic domain at a neutral pH [73,83,84,85,86,87,88,89,90,91,92,93,94,95]. Thus, legumain is shown to exhibit AEP, ACP, PAL, and aspartyl-peptidase activity, depending on the environment. The exact mechanisms regulating the shift between these different activities of legumain are currently not fully characterized.

Our understanding of the roles of legumain in the regulation of tissue homeostasis and pathogenesis of different diseases has gradually been expanded over the past three decades. In this review, we describe the current knowledge about the role of legumain in maintaining normal body functions and in pathologic processes involved in various diseases such as cardiovascular and cerebrovascular diseases, cancers, fibrosis, and neurodegenerative diseases, as well as aging and senescence (Figure 2). In addition, we also provide an overview of the approved drugs that have been shown to inhibit the AEP activity of legumain.

## 2. Role of Legumain in Kidney Homeostasis

Legumain is highly expressed in the late endosomes and lysosomes of the proximal tubular cells in the kidneys, where it plays a role in the processing of cathepsins (e.g., cathepsin B, H, and L) [37] and also contributes to control ECM remodeling through the degradation of different ECM proteins such as fibronectin or collagen [37]. Legumain also contributes to the degradation of some macromolecules (e.g., albumin) that get endocytosed by the renal proximal tubular cells and are destined to be degraded [37]. Since legumain is highly expressed in the kidneys, the involvement of legumain in regulating kidney homeostasis was one of the first investigations carried out using legumain-deficient mice. Legumain-deficient (*Lgmn* knockout) mice indicate normal fertility and viability, with no overt behavioral abnormality [83]. However, legumain deficiency leads to enlargement of the lysosomes in renal proximal tubular cells due to the accumulation of legumain substrate proteins [37]. Furthermore, processing of the lysosomal cysteine cathepsin B, H, and L was impaired in legumain-deficient mice [37]. Although accumulation of legumain substrates was observed in the kidney endosomes and lysosomes of legumain-deficient mice, no major defects in lysosomal proteolytic function were observed. However, histological abnormalities in the kidneys of legumain-deficient mice have been reported, such as hyperplasia of the proximal tubular cells, interstitial fibrosis, glomerular cysts formation, and renal pelvis dilation [23]. In addition, kidney function analysis indicated decreased glomerular filtration rate, increased plasma creatinine levels and proteinuria in legumain-deficient mice. These changes indicate that legumain has a role in the endosomal/lysosomal homeostasis of kidney proximal tubule cells [37] and that legumain deficiency causes altered kidney morphology and disturbed function [23]. 

## 3. Role of Legumain in Regulation of Hematopoietic Homeostasis

Legumain is involved in regulating the function of the hematopoietic system at multiple levels.

Approximately two decades ago, legumain was shown to be involved in processing of the invariant chain (Ii) and the proteolysis of antigens relevant for major histocompatibility complex (MHC) class II presentation in B cells [45]. In this regard, recognition of MHCII antigens by T cells is shown to be decreased in primary acute myeloid leukemia due to the downregulation of legumain expression and activity [84]. However, no major difference was found in Ii processing or maturation of MHC class II products in bone-marrow-derived dendritic cells or splenocytes isolated from legumain-deficient mice [85]. This controversy may be due to the variations in protease activities across different cell types.

Legumain is also shown to be responsible for the activation of cathepsin L in bone-marrow-derived dendritic cells, as the activated (two-chain) form of cathepsin L was not detected in these cells [85]. However, impaired cathepsin L processing did not cause major impacts on CD4+ and NK T cell numbers [85].

Legumain contributes to antigen presentation by the direct cleavage of the antigens [86] or indirectly through the activation of other proteases [87]. In this regard, it is shown that legumain contributes to the generation of microbial tetanus toxin cleavage products which subsequently are coupled with MHC class II molecules and presented to T cells [86]. However, investigating the immune response of legumain-deficient mice to tetanus toxin C fragments indicated that the initial T cell response was accelerated by the presence of legumain, whereas legumain deficiency only made a minor difference to the final antitetanus immune response [88]. These findings suggest the presence of in vivo compensatory mechanisms for tetanus toxin processing in the absence of legumain. 

Legumain is also involved in the processing and activation of Toll-like receptors (TLRs). TLRs are found in macrophages and dendritic cells and are involved in innate immunity and the recognition of microbial molecules. TLR7 and 9 are legumain substrates (Table 1) [58,61], and thus legumain is involved in diverse functions of the immune system that are dependent on TLR7 and/or 9 signaling, including the immune response to various viral infections.

Expression of legumain is highly induced during monocyte-to-macrophage differentiation [69], and after macrophage polarization, proinflammatory M1 secrete significantly more legumain than proresolving M2 macrophages [73]. Macrophages also secrete EVs containing the *LGMNP1* pseudogene, which is shown to be functional by upregulating legumain expression [89]. Surprisingly, the treatment of primary peripheral blood mononuclear cells with recombinant legumain during differentiation induced an anti-inflammatory phenotype by upregulation of anti-inflammatory and the downregulation of proinflammatory mediators [90]. These studies provide evidence for a possible role of extracellular legumain in regulation of immune responses.

Legumain is also involved in maintaining the integrity of the plasma membrane in red blood cells (RBC)/erythrocytes. Maturation of reticulocytes to erythrocytes includes remodeling of the plasma membrane in a process that is dependent of lysosomal function. The impaired lysosomal function due to legumain deficiency causes abnormal membrane protein destruction during erythrocyte development, leading to formation of defective erythrocytes that get engulfed by the histiocytes in the spleen and bone marrow [83]. Therefore, legumain-deficient mice have abnormally enlarged histiocytes with ingested RBC, together with reduced RBC count and age-dependent decline in hematocrit values. In addition, these mice develop a phenotype resembling the primary clinical symptoms of hemophagocytic syndrome, whereas there is no aberrant death in early age or significant elevation of circulating cytokines in these mice. Legumain-deficient mice were also reported to have elevated body temperature, cytopenia, and enlarged spleen with active extramedullary hematopoiesis.

Taken together, these studies indicate that legumain has an important role in regulating the homeostasis and function of the hematopoietic system.

## 4. Role of Legumain in Bone Remodeling

Bone remodeling is a life-long process where old bones are removed (resorbed) from the skeleton, followed by formation of new bone. Bone remodeling is the key process involved in maintaining the integrity of the skeleton through regeneration of microfractures, as well as adapting the skeleton to the mechanical needs of the body [91]. Bone resorption is mediated by osteoclasts that originate from hematopoietic stem cells, whereas the formation of the new bone is mediated by osteoblasts originating from bone marrow mesenchymal stromal cells (BMSCs). Legumain has been implicated in regulating the differentiation and function of both osteoclasts and osteoblasts.

It has been shown that legumain inhibits the formation of osteoclast-like multinucleated cells in human or mouse bone marrow cell cultures in the presence of 1,25-dihydroxyvitamin D3 (1,25-(OH)2D3) and parathyroid hormone-related protein (PTHrP) [92]. In addition, implantation of legumain-overexpressing HEK293 cells in immune-deficient mice decreased PTHrP-induced hypercalcemia, associated with decreased osteoclast surface and numbers [92]. In line with these findings, proteomic analysis of secreted proteins in cultures of RAW264.7 macrophages in the presence of RANK ligand indicated reduced legumain expression during differentiation to osteoclast-like cells [66]. Further, it is also shown that the inhibitory effect of legumain on osteoclast formation is mediated through its enzymatically inactive carboxy-terminal fragment, whereas proteolytically active legumain does not affect osteoclastogenesis [93]. 

Myeloid-derived suppressor cells (MDSCs) are immature immune cells that are involved in metastasis of e.g., breast cancer and have the ability to form osteoclast-like cells [94]. It is shown that inhibition of legumain enhances formation of osteoclast-like cells in MDSC cultures, possibly through altering the cleavage and activation of cathepsin L [94].

In addition to a role in regulation of osteoclast formation, legumain is involved in regulation of osteoblast differentiation and function. We have shown that legumain is expressed by BMSCs and inhibits differentiation towards the osteoblast lineage through its role in ECM remodeling by fibronectin degradation [41]. We have also shown that short hairpin RNA against legumain decreased legumain expression in BMSC cultures and enhanced in vivo bone formation capacity of the cells. Furthermore, we also showed that legumain-deficient zebrafish exhibited precocious formation of mineralized vertebrae. In contrast, legumain expression was increased during differentiation of BMSCs towards adipogenic lineage and presence of legumain in bone marrow adipocytes was inversely correlated with adjacent trabecular bone area in a cohort of patients with postmenopausal osteoporosis [41]. In line with these findings, it was recently shown that the activation of the brain-derived neurotrophic factor (BDNF)/TrkB signaling pathway prevents ovariectomy-induced bone loss in mice through the activation of Akt signaling, inhibition of legumain, and increased osteoprotegerin levels [95].

## 5. Role of Legumain in Vascular Homeostasis, Cardiovascular, and Cerebrovascular Diseases

Legumain has been shown to be involved in the regulation of vascular homeostasis and the related pathologies (especially in heart and brain tissues) due to its diverse functions such as roles in ECM remodeling, the activation of other proteases (e.g., MMPs), and the regulation of immune cell functions.

Cardiovascular diseases (CVDs) include various diseases of the heart and/or blood vessels and are leading causes of death worldwide, with coronary artery disease (CAD) accounting for 16% of total deaths [96,97]. A major cause of CVDs is atherosclerosis, where lesions (plaques) grow within coronary arterial walls and limit the blood supply to the heart muscle [98]. Atherosclerosis is characterized with low-grade inflammation in the vessel wall of large and medium-sized arteries, and with subsequently lipid accumulation. As the lesion develops, the vessel lumen is narrowed and subsequently blocks the heart blood supply, leading to acute coronary syndrome (ACS) including myocardial infarction (MI). If the plaque ruptures, cerebrovascular diseases may occur due to blockage of smaller arteries in the brain causing stroke. 

The pathological role of MMPs in the development of atherosclerotic lesions and plaque rupture are extensively studied and recently reviewed [99,100]. However, the role of cysteine proteases and especially legumain are only starting to be elucidated. Several studies have highlighted the role of legumain in CVDs and particularly in the formation and vulnerability of atherosclerotic plaques. Nearly 15 years ago, legumain was shown to be highly upregulated in unstable carotid plaques, where unstable regions of the plaques contained twice the amount of active legumain compared to the stable regions, suggesting the possible contribution of legumain to plaque instability [101,102]. Expression of legumain in vascular lesions indicates possible implications in atherogenesis [101] and legumain is one of the 18 genes associated with atherosclerotic plaque rupturing [103]. We and others have found increased legumain in atherosclerotic plaques colocalized with macrophages [73,104]. Moreover, legumain is present in blood platelets and released upon platelet activation [90].

Elevated legumain is not only limited to the atherosclerotic plaques but also found in the circulation, making legumain a potential CVD biomarker of great interest. Elevated circulatory legumain is measured in various CVD patients, including patients with carotid atherosclerosis [73], peripheral artery disease [105], pulmonary arterial hypertension [106], complex coronary lesions [74], acute myocardial infarction [107], and thoracic aortic dissection [79], as compared to healthy controls. In addition, legumain is linked to aortic aneurysm [108].

In addition, it is also shown that legumain is involved in vascular degeneration, dissection, and rupture, thereby playing role in the pathogenesis of thoracic aortic dissection (TAD) [79]. Legumain expression was shown to be increased in the aorta from TAD patients, as well as a murine preclinical model of TAD. Genetic deletion of *Lgmn* or pharmacological inhibitin the AEP activity of legumain was shown to ameliorate TAD symptoms in the preclinical murine model. The effect of legumain in TAD pathogenesis is suggested to be mediated through binding to integrin αvβ3 in vascular smooth muscle cells (VSMCs) and subsequent inhibition of Rho GTPase activity in these cells, compromising VSMC differentiation [79].

The potential role of circulating legumain levels as a prognostic CVD biomarker has been evaluated in several studies. We have recently shown that the circulating levels of legumain are increased during acute cardiovascular events and are associated with improved outcome after 3 years [90]. In a large patient population with ACS (*n* = 4883), increased circulating legumain at baseline was associated with the composite end-point (cardiovascular death, spontaneous myocardial infarction, or stroke) [109]. However, this association was not clear when adjusted for biomarkers including C-reactive protein, troponin T, cystatin C, growth/differentiation factor 15, and N-terminal probrain natriuretic peptide. Interestingly, high levels of legumain one month after ACS was negatively associated with the occurrence of stroke. These findings illustrate the complex and dual role of legumain during ACS and atherogenesis. Furthermore, in acute MI, circulating levels of legumain is shown to be a predictor of all-cause mortality, also after adjusting for clinical confounders [107]. In patients with type 2 diabetes mellitus, high serum legumain is significantly correlated with increased risk of peripheral artery disease [105].

Whether increased circulating legumain levels in CVD is beneficial or not is still unclear [110]. There are several postulated mechanisms of legumain action in CVDs. Legumain is shown to regulate oxidized LDL-induced macrophage apoptosis through enhancement of an autophagy pathway involved in atherogenesis and formation of atherosclerotic plaques [111]. Moreover, we have shown that legumain is mediating the anti-inflammatory effects on macrophages [90], a novel and contradictory role of legumain versus the predicted role in the plaque destabilization mentioned above. Legumain downregulates the expression of M1 macrophage markers but upregulates M2 markers, indicating stimulation of a shift towards an anti-inflammatory, proresolving macrophage phenotype. Legumain also has a positive effect in improving cardiac repair after MI by promoting clearance and degradation of apoptotic cardiomyocytes [112]. On the other hand, in apolipoprotein-E-deficient mice, legumain can contribute to the development of atherosclerotic lesions and induction of atherosclerotic vascular remodeling by promoting ECM degradation and migration of monocytes and vascular smooth muscle cells [113]. In pulmonary artery hypertension, macrophage-derived legumain is involved in the disease development by inducing MMP-2/transforming growth factor β1 (TGF-β1) signaling [106]. Furthermore, legumain may promote ECM degradation due to the ability of activating proMMP-2 [49], processing of cysteine cathepsins [23,37], or by the direct proteolysis of ECM components such as fibronectin [40]. 

Cerebrovascular diseases such as stroke are also among major causes of death worldwide, accounting for 11% of total deaths [96]. Legumain is also involved in cerebrovascular diseases such as stroke and cerebral hypoperfusion. Increased legumain expression in the peri-infarct area was observed after transient occlusion of the middle cerebral artery; however, legumain was not essential for the functional deficit in a rat stroke model [72]. In a mouse model mimicking chronic cerebral hypoperfusion, the depletion of legumain improved cognitive impairment by reducing neuroinflammation, as legumain is upregulated and triggers synaptic plasticity impairment and neuroinflammation [114]. In newly diagnosed progressive ischemic stroke patients, circulatory legumain was significantly higher than in patients with nonprogressive ischemic stroke but was not correlated with blood pressure variability and neurological outcome [115]. In addition, it is shown that legumain contributes to the immune responses during stroke by indirectly altering the number of CD74+ cells in the ischemic hemisphere by the modification of molecules involved in immune cell attraction [72].

## 6. Role of Legumain in Fibrosis

Fibrosis is characterized as a pathological wound-healing process where fibrous connective tissue replaces the destroyed normal tissues, leading to defective tissue remodeling, the formation of permanent scar tissue, and the loss of normal tissue functions. Fibrosis with the dysregulated production of ECM proteins, especially collagen and fibronectin, occurs in different tissues such as the kidneys, lungs, and pancreas, and is often associated with increased risk of cancer in these organs. TGF-β1 is an important profibrogenic factor in organs with extensive fibrogenesis and profibrogenic factors are amplified after an injury [116]. Several studies have indicated a role of legumain in fibrogenesis, either by the processing of ECM proteins or by indirectly activating TGF-β1 via processing of proMMP-2. 

As mentioned above, legumain is required for normal kidney physiology and homeostasis, and legumain-deficient mice, as well as tubule-specific legumain-deficient mice from 3 months of age, develop and exhibit renal interstitial fibrosis [23,117]. The development of renal interstitial fibrosis is probably due to the role of legumain in ECM remodeling through degradation of fibronectin in renal proximal tubular cells [40]. Additionally, more severe fibrotic lesions are observed in a unilateral ureteral obstruction model in legumain-deficient compared to wild-type mice [70]. Additionally, the loss of legumain mediates renal fibrosis as legumain deficiency induces premature senescence and the secretion of profibrotic cytokines during age-related renal fibrotic injury [117]. However, legumain deficiency not only mediates fibrogenesis, but legumain has been shown to attenuate renal interstitial fibrosis in obstructive nephropathy, possibly by mediating effects on M2 macrophages [70]. In contrast, legumain is shown to promote fibrogenesis in chronic pancreatitis, and it is suggested that macrophage M2-derived legumain mediates activation of pancreatic stellate cells and increases the synthesis of ECM proteins via the activation of TGF-β1 [118]. Legumain deficiency or inhibition reduces the severity of pancreatic fibrosis by suppressing activation of the TGF-β1 precursor. In addition, in patients with idiopathic pulmonary fibrosis (IPF), the level of circulating legumain is suggested as one of the six biomarkers in an IPF clinical decision index, since legumain is involved in promoting disease progression [119]. These studies indicate that the role of legumain in fibrosis is highly tissue-dependent.

## 7. Role of Legumain in Age-Related Tissue Senescence

Aging is an independent risk factor for many disorders and is associated with impaired cell and tissue functions, leading to organ dysfunction and disease. The age-dependent changes of cell and tissue functions occur due to intrinsic and extrinsic factors, such as accumulation of DNA damage, telomere shortening, and cellular senescence, as well as diverse environmental insults, etc. [120]. In addition, there is increasing evidence that impaired lysosomal function contributes to control the aging process [121,122]. 

Being among the key proteases involved in lysosomal protein degradation, legumain has been examined in the context of aging. It is shown that legumain expression and activity are significantly increased in the brain during aging and may contribute to the pathogenesis of age-related neurodegenerative disorders (see below) [30,56]. However, our studies of legumain levels in human sera indicated that the circulating levels of legumain decrease by aging in patents with postmenopausal osteoporosis [41]. As kidneys have the highest level of legumain expression in the body, we speculated that the age-related impairment of kidney function could involve increased legumain excretion or decreased expression and thus account for the age-related decrease in circulating legumain levels. Interestingly, Wang et al., recently reported that legumain expression in the kidneys is significantly decreased during aging in mice [117]. This study also showed that legumain-deficiency leads to age-related renal fibrosis due to impaired mitophagy and accumulation of defective mitochondria. This in turn leads to elevated ROS production and a premature senescence kidney phenotype, at least at the histological level [117]. However, preclinical studies in aging mice have shown that pharmacological inhibition of legumain does not lead to adverse effects on the kidney, liver, heart, spleen, or brain [55]. Furthermore, the above-mentioned studies indicate that the circulating levels of legumain are not necessarily an indicator of legumain expression in different tissues. Therefore, the role of legumain in regulation of homeostasis in different tissues during aging and possible contribution to pathogenesis of other age-related disorders has yet to be investigated.

## 8. Role of Legumain in Neurodegenerative Diseases

The growing number of people suffering from neurodegenerative disorders such as Alzheimer’s disease (AD), Parkinson’s disease (PD), amyotrophic lateral sclerosis (ALS), and multiple sclerosis (MS) is a major global issue. A common feature of several neurodegenerative disorders is the accumulation of proteins which are either misfolded or cleaved to form neurotoxic aggregates that play roles in the disease pathogenesis. Legumain (δ-secretase) is shown to cleave a number of substrates in the brain (Table 1) which are important in the pathogenesis of neurodegenerative disorders. Thus, legumain is an interesting drug target or disease biomarker in neurodegenerative diseases (reviewed in [123,124]).

Amyloid precursor protein (APP) and Tau are pathological hallmarks of Alzheimer’s disease (AD), and both proteins are legumain substrates (Table 1). APP cleavage by legumain promotes the production of amyloid β (Aβ) (reviewed in [125]). Legumain cleavage of Tau promotes Tau phosphorylation and thus abolishes microtubule assembly [56]. The serine–arginine protein kinase 2 (SRPK2) mediates Tau phosphorylation and pre-mRNA splicing in neurons, subsequently suppressing Tau-dependent microtubule polymerization and the axonal elongation of neurons [126]. SRPK2 is also cleaved by legumain (Table 1), contributing to increased kinase activity and accelerated cognitive decline in tauopathies due to nuclear translocation, mediating Tau-splicing imbalance [51]. In addition, legumain cleaves amphiphysin I and the DNase inhibitor SET/inhibitor 2 of protein phosphatase 2 (I2PP2A; Table 1), mediating Tau hyperphosphorylation and synaptic dysfunction [32] or neuronal cell death [44]. These phenomena are increased during aging due to increased legumain activity. Brain ischemia and hypoxia generate brain acidosis, which subsequently activate legumain to cleave I2PP2A leading to Tau hyperphosphorylation, suggesting the involvement of brain acidosis in AD pathology [43]. The ischemia-induced brain injury was ameliorated in legumain-deficient mice challenged with brain acidosis, as compared to normal (wild type) mice, providing evidence for a role of legumain in the pathogenesis of ischemia-induced brain injury [43]. Furthermore, substantially less Aβ-deposition reduced Tau phosphorylation and pathological changes have been observed in legumain-deficient mice [33,56]. A disintegrin and metalloproteinase 10 (ADAM10), as well as ADAM17, cleave APP and thereby avoid Aβ production. Both ADAM10 and 17 are legumain substrates (Table 1) and are cleaved and degraded by legumain in the lysosomes [28], which subsequently could trigger AD onset or progression. In addition, legumain has been suggested as an AD prognostic biomarker and elevated legumain monitored by live animal imaging was shown to be an early sign of AD onset in an AD pre-clinical murine model [127].

The therapeutic targeting of legumain has been proposed as a novel strategy in the treatment of AD, as legumain deficiency is shown to inhibit neuroinflammation and protect against Aβ-induced cognitive deficits and synaptic plasticity dysfunction [128]. A few legumain inhibitors have been studied in the treatment of AD in mice models. Compound 11 was identified using high-throughput screening approach and shown to inhibit legumain allosterically by a dual active site and subsequently reduced APP and Tau cleavage [55]. Furthermore, the administration of compound 11 to senescence-accelerated mice decreased legumain activity, Aβ generation, Tau fragmentation, and hyperphosphorylation in the brain, as well as memory loss [129]. Moreover, an agonist of the BDNF receptor has been shown to inhibit legumain activity in the brain of AD mice, acting synergistically with compound 11 [130]. Furthermore, the irreversible peptide inhibitor RR-11a was very recently shown to be efficiently delivered intracerebrally by nanobubbles and shown to improve cognitive function and inhibit APP and Tau cleavage by legumain in an AD mouse model [131]. Interestingly, gut inflammation seems to initiate AD-associated pathologies mediated by the vagus nerve to the brain and trigger AD pathology and cognitive dysfunction [132]. In addition, legumain is shown to inhibit axon regeneration in peripheral nerves [133].

Legumain has been suggested to also have a key role in the pathogenesis of Parkinson’s disease (PD) due to the cleavage of α-synuclein (Table 1), triggering the aggregation of the cleavage products in Lewy bodies, the neurotoxicity of dopaminergic neurons, and movement disorders [30]. Inhibition of legumain was shown to protect against neurodegeneration by a pesticide known to induce PD via suppression of α-synuclein aggregation and neuroinflammation [134]. The triggering factor of PD is still unknown but initiation is suggested to be mediated by legumain through cleavage and aggregation of α-synuclein in the enteric nervous system in the gut which spreads to the central nervous system [135]. Abnormal metabolism of dopamine by monoamine oxidase B (MAO-B) has been shown to contribute to dopaminergic neurodegeneration in PD. Moreover, α-synuclein is shown to directly bind to and enhance the activity of MAO-B, which subsequently increases legumain activity and α-synuclein cleavage, mediating PD pathology [35]. Synaptojanin 1 is another legumain substrate (Table 1) and its cleavage triggers synaptic dysfunction in PD [52]. As mentioned, TLR9 is a legumain substrate (Table 1) and TLR9 activation via microglial glucocorticoid receptors contributes to degeneration of dopaminergic neurons in the brain [136]. Legumain deficiency in a mouse PD model induced by MPTP (1-metyl-4-fenyl-1,2,3,6-tetrahydropyridin) was recently shown to improve cognitive impairments by inhibiting proinflammatory microglial activation [137]. These studies provide preclinical evidence that targeting of legumain could be a promising novel approach in the treatment of PD.

Aggregates of TAR DNA-binding protein (TDP-43) cleavage products are found in subtypes of frontotemporal dementia and amyotrophic lateral sclerosis (ALS). TDP-43 is another substrate of legumain (Table 1) [53] and a nuclear protein involved in RNA splicing. During development of multiple sclerosis (MS), the protective myelin sheaths of neurons are attacked and destroyed by the immune system, which exerts neurotoxic effects and also decreases the interneuron communication (reviewed in [138]). Myelin basic protein (MBP) is an important structural element of myelin and is shown to be a legumain substrate (Table 1) when MBP was incubated with purified legumain [47]. However, it is shown that lysosomal processing of MBP in human B lymphocytes is mediated by cathepsin G and not legumain [139]. In two experimental MS murine models, the upregulation of legumain in association with increased lesion activity was documented by proteomic analyses [140]. Furthermore, anxiety- and depressive-like behaviors are reduced and abilities of spatial cognition are improved in legumain-deficient mice [141].

## 9. Role of Legumain in Cancer

For more than 20 years, legumain has been linked to the malignancy and poor prognosis of several cancer types and was recently reviewed by Zhang and Lin [142]. Herein, recent and additional information regarding legumain in cancer will mainly be covered. The overexpression of legumain was initially shown in various solid tumors, as well as in lymphoma and melanoma [75]. For example, in breast cancer, legumain is expressed in tissue-associated macrophages (TAMs) as well as in stromal cells, and its circulating levels are significantly increased, leading to the promotion of cancer-cell proliferation and metastasis [143]. In addition, elevated legumain expression levels are considered a negative prognostic marker [144]. In contrast, the endogenous legumain inhibitor cystatin E/M is considered a breast cancer suppressor [145]. In glioblastoma, legumain has recently been shown to cleave the tumor suppressor p53 (Table 1), mediating p53 inactivation and subsequently cellular and genetic destabilization [48]. Legumain was also demonstrated to cleave tropomodulin-3 (Tmod3; Table 1), a ubiquitously expressed regulatory protein of the cytoskeleton [62]. Tmod3 cleavage by legumain produces functional truncated Tmod3 which is detected in various tumors and associated with poor prognosis of high-grade glioma.

The tumor microenvironment is known to be hypoxic and plays an important role in cancer progression and metastasis. In multiple myeloma, a hematological malignancy in the bone marrow, legumain was one of the nine proteins regulated by chronic hypoxia [146]. Moreover, legumain is involved in the production of oxygen reactive species and metastasis in breast cancer, which is counteracted by legumain deficiency or inhibition [147]. 

The legumain pseudogene (*LGMNP1*) is overexpressed in glioblastoma cells resistant to radiotherapy [148] and was shown to be functional in promoting tumor progression by acting as a microRNA (miR-495-3p) [26]. The miR-495-3p is a short-stranded RNA blocking post-transcriptional translation of legumain [149] and is proposed as a biomarker in some cancer types. Furthermore, *LGMNP*1 also promotes thyroid carcinoma progression via the miR-495 and autophagy pathway [150].

Glioblastoma cells secrete EVs containing active legumain to the tumor microenvironment [48]. The *LGMNP1* pseudogene from the EVs of ectopic endometrial stromal cells induces M2-like macrophage polarization by upregulating legumain and was recently shown to be a promising predictive biomarker for ovarian endometriosis recurrence [89]. In epithelial ovarian cancer, legumain was found in EVs colocalized with integrin αvβ1 and was considered to be crucial for peritoneal metastasis [151].

Targeting legumain is now a new strategy in cancer therapy (reviewed by [152,153]). In addition, several legumain-activated prodrugs have been constructed for various cytotoxic drugs, including doxorubicin, etoposide, auristatin, and colchicine [75,154,155,156]. The strategies and challenges for such protease-activated prodrugs containing a legumain-cleavable peptide has recently been reviewed [157]. 

Recently, a lytic peptide conjugated to paclitaxel was shown to have a dual and synergistic cytotoxic effect after cleavage by legumain in the tumor [158]. The first legumain-based DNA vaccine was developed in 2006 and was shown to induce a marked CD8+ T cell response by reducing tumor TAMs density and their release of proangiogenic factors, and subsequently the suppression of tumor growth, angiogenesis, and metastasis [143]. The proton pump inhibitor esomeprazole is an approved drug that is shown to prevent cancer cell invasion in vitro and metastasis in vivo by inhibiting the AEP activity of legumain [159]. 

## 10. Pharmacological Inhibition of Legumain

Over the years, a number of small-molecule legumain inhibitors have been developed and are reviewed elsewhere [160,161]. Discovering the role of legumain in the pathogenesis of various diseases, as discussed above, has initiated an approach of targeting legumain or utilizing legumain for prodrug activation, which may provide novel therapeutic opportunities for, e.g., cancer therapy [152,153]. However, to our knowledge, so far there has not been any clinical trials involving legumain inhibitors.

We and others have shown that various approved and marketed drugs such as statins (3-hydroxy-3-methylglutaryl coenzyme A (HMG-CoA) reductase inhibitors) [80] are able to inhibit the AEP activity of legumain. We have shown that cell treatments with either simvastatin or atorvastatin reduced legumain activity in skeletal muscle cells or macrophages, respectively [69,162]. In addition, atorvastatin has been shown to downregulate legumain mRNA in monocytes/macrophages isolated from atherosclerotic patients [163]. 

Statins are drugs commonly used to lower the blood cholesterol level by inhibiting HMG-CoA reductase in the cholesterol synthesis pathway. In addition, statins have been shown to have pleiotropic effects, including anti-inflammatory and immunomodulatory properties. Statins have also been shown to improve bone formation and have been suggested to be used for the treatment of bone loss diseases [164,165]. As legumain is shown to inhibit bone formation [41], it is possible that legumain inhibition plays role in mediating the bone formation enhancing effect of statins. In addition, statins reduce the risk of Parkinson’s disease [166,167,168], Alzheimer’s disease, and related dementias [169,170,171]. Furthermore, statins have been shown to slow the deterioration of neuropsychiatric status and exert beneficial effects on the short term scores of Mini-Mental State Examination scale, and significantly improve the daily living ability in Alzheimer’s disease patients [172]. Given the role of legumain in the pathogenesis of Alzheimer’s disease and Parkinson’s disease, it is possible that legumain inhibition contributes to the beneficial effects of statins in these neurodegenerative diseases. 

Proton pump inhibitors (PPIs) are commonly used in the treatment of peptic ulcer diseases [159,173] and have been also suggested to have an anticancer effect [174], whereas their therapeutic benefits for cancer patients is not yet established [175]. We have demonstrated that the PPI lansoprazole directly inhibits legumain by binding to the cysteine in the active site [173], which has been confirmed for esomeprazole and omeprazole [159]. Currently, it is not clear whether legumain inhibition by PPIs has any role in mediating the anticancer effect of these drugs. Thus, the exact mechanisms and consequences of legumain inhibition by statins or PPIs still remain to be elucidated.

## 11. Conclusions

This review provides an update on our understanding of legumain, a unique cysteine protease with nonredundant asparaginyl endopeptidase activity. Our understanding of the legumain substrate targets and its mode-of-action in the regulation of different biological processes has increased over the last years. Current data suggest that legumain is not only involved in regulating renal, hematopoietic, skeletal, and vascular physiology, but also has roles in the pathogenesis of diverse malignant and nonmalignant diseases. Legumain has been proposed as a novel therapeutic target for the treatment of neurodegenerative diseases such as Alzheimer’s and Parkinson’s disease, osteoporosis, and different cancers. In this regard, it is important to highlight that statins inhibit the proteolytic activity of legumain and the observed therapeutic benefits of these widely used drugs in Alzheimer’s disease, Parkinson’s disease, osteoporosis, and different cancers could be related to the inhibition of legumain. Additional studies are warranted to investigate this hypothesis. Providing evidence of legumain inhibition in mediating these beneficial therapeutic effects of statins can provide valuable clinical information about the safety of legumain-targeting and expedite clinical application of legumain inhibitors. In addition to the direct inhibition as a therapeutic target, legumain can also be employed for prodrug activation that could be of particular interest for, e.g., cancer treatment. Additional preclinical and clinical studies are needed for the further development of this approach.

## Figures and Tables

**Figure 1 ijms-23-15983-f001:**
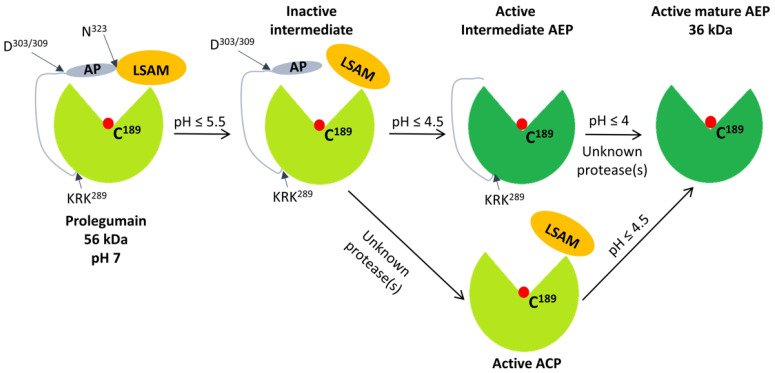
Illustration of legumain activation. Inactive prolegumain (56 kDa) is composed of the catalytic domain (light green), activation peptide (AP, grey), and legumain stabilization and activity modulation domain (LSAM, orange), and is stable at a neutral pH. The proenzyme undergoes autocatalytic processing at asparagine 323 (N323) at pH ≤ 5.5 and at aspartate 303/309 (D303/309) at pH ≤ 4.5. Another in trans processing occurs after the KRK289 motif by unknown protease(s). Release of the LSAM domain is mandatory to gain AEP activity (dark green) and the cysteine (C189) in the catalytic site is marked (red dot). In addition, ACP activity is obtained when the LSAM remains electrostatically bound, whereas the AP is removed. A decrease in pH causes protonation and release of the LSAM domain and legumain acquires AEP activity. The amino-terminal cleavage after aspartate 21 (D21) and 25 (D25) is not shown. AEP, aspaparaginyl endopeptidase; ACP, asparaginyl carboxypeptidase. Created by PowerPoint.

**Figure 2 ijms-23-15983-f002:**
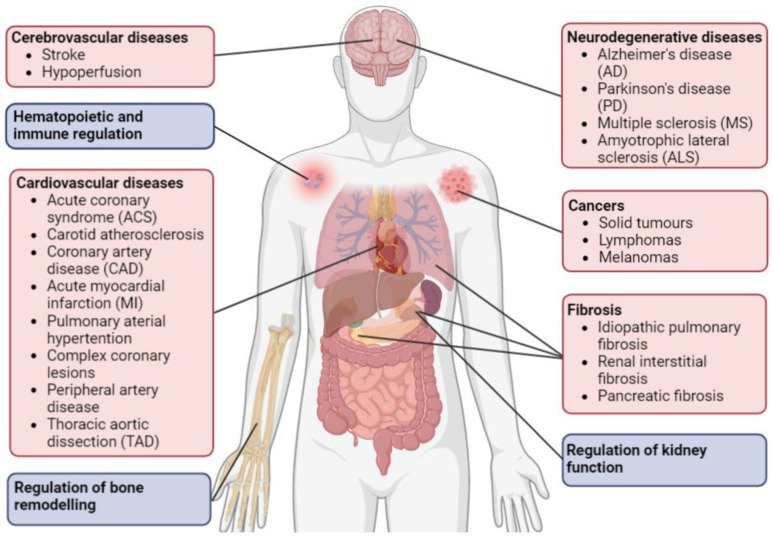
Illustration of the involvement of legumain in various physiological processes (blue boxes) and diseases (red boxes) in different organs and tissues of the body. See the main text for description and references. Created by BioRender.

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
