# Peer review of "The Mammalian Cysteine Protease Legumain in Health and Disease"

_ijms, 2022, doi:10.3390/ijms232415983_

Round 1
Reviewer 1 Report
The authors showed us an overview of cysteine protease legumain about its structure, activity activation, and described how the cysteine protease legumain played a crucial role in physiology and various diseases in detail. However, some questions need addressing before this review is ready for publication.
1. Why was the legumain mainly expressed in kidney? Were there other sites could express this protease?
2. The authors claimed that legumain was regulated by environment, such as pH value, various substrates and so on. Whether did these factors mediate legumain activity by the same mechanism? It will be great if the authors could map the possible pathways.
3. The structure of this review needed improvement. It would be great the authors described its role in physiological and pathological involvements separately. In other words, the function of legumain could be divided into two parts, including physiology and pathology. Moreover, the authors can also discuss about signaling and transcription factor of legumain in physiology.
4. The authors needed to add some sentences, which served as a connecting link between the preceding and the following, to make each part read smoothly and logically.
5. The authors needed to provide more evidence about application and development of legumain inhibitors in diseases. What were the perspectives of these inhibitors of legumain in diseases?
Author Response
Please see attachement.

Reviewer 2 Report
The review by Solberg et al. of insights into legumain functions in health and disease is very well organized and deserves to be published in the International Journal of Molecular Sciences. However, I would like the following points to be corrected.
1. Regarding Figure 1, the crystal structure of human prolegumain should also be shown to aid understanding at the molecular level.
2. With respect to Table 1, please provide a description of the activation, processing or degradation of substrates mediated by legumain.
3. At the end of the paper, they should add a conclusion or future perspective chapter.
Author Response
Please see attachement.

Round 2
Reviewer 1 Report
All my comments were replied clearly by the authors, and I had no further question.